# SPARSE HIERARCHICAL TABLE ENSEMBLE

## ABSTRACT

Deep learning for tabular data is drawing increasing attention, with recent work attempting to boost the accuracy of neuron-based networks. However, when computational capacity is low as in Internet of Things (IoT), drone, or Natural User Interface (NUI) application, such deep learning methods are deserted. We offer to enable deep learning capabilities using ferns (oblivious decision trees) instead of neurons, by constructing a *Sparse Hierarchical Table Ensemble* (S-HTE). S-HTE inference is dense at the beginning of the training process and becomes gradually sparse using an annealing mechanism, leading to an efficient final predictor. Unlike previous work with ferns, S-HTE learns useful internal representations, and it earns from increasing depth. Using a standard classification and regression benchmark, we show its accuracy is comparable to alternatives, while having an order of magnitude lower computational complexity. Our PyTorch implementation is available at `https://anonymous.4open.science/r/HTE_CTE-60EB/`.

## 1 INTRODUCTION

During the last decade, Deep Neural Networks (DNNs) have become dominant for many machine learning tasks in computer vision, natural language processing, speech recognition, and others (Goodfellow et al., 2016). However, when dealing with tabular data, DNNs superiority is in doubt with non-deep models such as Gradient Boosting Decision Tree (GBDT) (Friedman, 2001) being the top choice for many machine learning practitioners. Currently, deep learning and deep neural networks mostly refer to the same thing. Though the tight coupling between these two concepts is natural, given the numerous published studies, these are two different concepts. Deep learning is the learning of hierarchical useful representation, while neural networks are predictors using dot-product based neurons as their basic elements. In this work, we decouple the connection of deep learning to DNNs by introducing a deep learning alternative based on differentiable ferns as the basic computation unit. Ferns (also termed Oblivious Decision Tree (fern) in Popov et al. (2019); Kohavi (1994); Lou & Obukhov (2017)) are using $K$ simple feature comparisons to choose among $2^K$ possible outputs, and are hence far more expressive than a linear neuron with an activation function. In turn, deep predictors based on ferns can enjoy extreme sparsity, and provide accuracy-speed trade off far beyond standard DNNs.

We introduce *Sparse Hierarchical Table Ensemble* (S-HTE), specifically design to face tabular data tasks when extremely fast inference is needed. The basic neuron in a S-HTE architecture computes the answer of $K$ simple binary questions, creating a $K$-length binary word. This word is then used as an index to a table from which the required output is retrieved. The analog of a neuron layer (an ensemble of neurons), is an ensemble of $M$ tables, applied with their output summed. The deep S-HTE consists of several layers, constructed hierarchically in order to learn deep representations. During training, the architecture enables soft (ambiguous) splits in each fern node, allowing the examples to flow in each of the fern's branches with a certain probability. This enables the S-HTE to be completely differentiable and hence to be fully optimized using SGD. However, since we aim for extremely fast inference, we gradually sharpen the split decisions during the training process, so only one leaf is active in each fern at inference time.

S-HTE can be implemented using any of the recent deep learning frameworks (e.g. TensorFlow, Pytorch, etc.), and scales well to large problems, effectively trading speed for memory. While ferns have been used in classification before (Popov et al., 2019; Krupka et al., 2014), the method of Popov et al. (2019) is not sparse, while the ensembles of Krupka et al. (2014) are not deep. Our method is

hence the fern-based method enabling results with accuracy comparable to deep MLPs, yet with a fraction of the computing power.

The main motivation to deviate from dot-product neurons is the need for fast and efficient CPU inference. Ferns are a good choice since they have a much better relation of representation capacity to computational cost than a linear neuron. In representation richness terms, a binary classifier built from a single fern with $K$ queries has a VC-dimension of $O(2^K)$, compared to $O(K)$ for a linear neuron with $K$ inputs. However, computationally they have the same $O(K)$ cost. Considering the index produced the $K$ fern queries as a one-hot vector, this intermediate representation has $2^K$ dimensions, but is nevertheless $2^{-K}$-sparse. The table transforming the codeword to the next representation has $2^K$ rows, each containing an output vector for the next level D-dimensional representation, for a total of $2^K D_{out}$ parameters. Thus, a single fern provides a rich transformation family of $O(2^K D_{out})$ parameters, computed using just $O(K + D_{out})$ computations.

We demonstrate that despite the significantly non-linear elements and the extreme sparsity, networks based on S-HTE layers can be successfully trained end-to-end. On a recent benchmark of tabular data tasks (Gorishniy et al., 2021) the method achieves accuracy comparable or slightly lower than state of the art methods while using an order of magnitude lower computational complexity. This trade off is highly valuable for applications on low-end GPU-lacking devices prevalent for example in IOT, embedded or drone applications. Such low computational complexity capabilities are also required for 'keep-alive' devices, which demands maintaining long battery life.

Our main contribution is hence in introducing a fern-based method which is both deep and sparse, while previous similar methods only have one of these qualities. We show that this combination enables gaining order of magnitude in compute power, with small accuracy costs of $2 - 3\%$ at most on a contemporary benchmark. Our Pytorch code is open and available to the community [1].

The rest of the paper is organized as follows: in section 2 we discuss the most relevant recent work. In section 3 we describe our proposed S-HTE architecture. The computational complexity of the S-HTE and other competitors is analyzed in section 4. The experimental details are detailed in section 5, and section 6 contains some concluding remarks.

## 2 RELATED WORK

In recent years gradient-based deep learning models for tabular data is getting increasing attention. while our method mainly pinpoints inference speed, the vast majority of the latest work focuses on accuracy improvements (Kadra et al., 2021; Gorishniy et al., 2021; Popov et al., 2019; Katzir et al., 2020; Fiedler, 2021; Yang et al., 2018). In this section we review some of the latest gradient-based models, as well as other non-differentiable models which are the most relevant to the proposed method.

**Ensemble-based and multi-layered non-differentiable methods.** For most tabular data tasks, decision trees based architectures are currently the top-choice for both researchers and practitioners. These architectures are divided into two main groups - ensemble of decision trees, and multi-layered models with decision trees being the computational components. Ensemble of decision trees, mostly trained in a boosting like manner, includes several widely used implementations, such as XGBoost (Chen & Guestrin, 2016), LightGBM (Ke et al., 2017), and CatBoost (Prokhorenkova et al., 2017). Though these vary in details, their performance is quite similar. For example, the computational component in XGBoost are Random Decision Trees (RDTs), but in CatBoost these are ferns. Ferns are slightly weaker learners than trees in terms of capacity, but they are faster on modern machines with SIMD (Single Instruction Multiple Data) capabilities (see 3.1 for further discussion). Another line of work is stacking several non-differentiable layers into multi-layered architectures (Zhou & Feng, 2017; Miller et al., 2017). For example, Zhou & Feng (2017) presented a deep random forest architecture in which several layers are stacked. Since random forest are not suitable for gradient based learning, each such layer is trained separately, lacking the capability for end-to-end training.

**Differentiable decision trees based architectures.** The above methods are widely used and achieves state of the art results on tabular datasets, but do not enable gradient flow and end-to-end representation learning. To address this issue, several works enabling gradient based learning with

---

[1]https://anonymous.4open.science/r/HTE_CTE-60EB/

decision trees were published in recent years (Yang et al., 2018; Kontschieder et al., 2015; Hazimeh et al., 2020; Popov et al., 2019). The authors in Kontschieder et al. (2015) proposed the deep neural decision forest, in which feature representation is learned using CNNs, and the classification is done using differential decision trees. Hazimeh et al. (2020) propose to create differentiable tree routing, by smoothing the decision functions in the internal tree nodes. Popov et al. (2019) suggested the NODE architecture, in which they smooth ferns into differentiable ferns using the entmax function (Peters et al., 2019), but the fern output is dense with all the $2^K$ fern leaves active at both train and test time. The output of each layer is a logit classification vector, which is concatenated with the input vector and pushed to the next layer in the architecture. The final prediction logit (2 or 3 output neurons in the experiments) is the average across all layer predictions. Flat and deep architectures ({2,4,8} layers) were tested, with 2048 ferns each. Since the output neurons of each layer are direct classification logits (Gorishniy et al., 2021) there is no learning of internal representation. In a recent study (Kadra et al., 2021) this method was found non scalable due to memory constraints or run time issues (above 4 days of training), resulting from the unlimited dense output of the ferns involved. The S-HTE method does not suffer from these problems. It learn an internal representation which is not directly class-related. To achieve scalability and efficient inference it structurally limits the number of active fern leaves, and shifts toward sharp ferns as training progresses.

**State of the art DNNs for tabular data.** There is an extensive research on how to operate standard DNN models on tabular data. For example, Kadra et al. (2021) reports that using a standard MLP (with or without dropout) achieves inferior results. However, by using several regularization techniques, they were able to boost the MLP performance. Gorishniy et al. (2021) showed that using a standard ResNet provides state-of-the-art results on several known datasets. Shavitt & Segal (2018) propose to infuse feature importance to standard DNNs by adding a coefficient for each feature weight and solving the task using hyperparameter tuning scheme (resulting with what they termed - 'regularization learning networks'). However, in their paper, they do not compare with properly tuned GBDT implementations, which are the most appropriate baselines. Another line of work is to incorporate attention mechanisms to solve tabular datasets tasks(Song et al., 2019; Arık & Pfister, 2020; Gorishniy et al., 2021). FT-Transformer (Gorishniy et al., 2021) is such an example, using a *feature tokenizer* to embed the input features and then applying several stacked attention layer over the embedded features.

## 3 METHOD

In this section we introduce the core building block in our algorithm, then move on to explain how to train such a model using gradient based learning

### 3.1 INTRODUCTION TO FERNS

The basic computing element of the S-HTE model accepts a representation vector $\boldsymbol{x} \in \mathbb{R}^{D_{in}}$ and outputs a representation vector $\boldsymbol{y} \in \mathbb{R}^{D_{out}}$. The transformation is a pair of a word calculator ($W$) and a voting table ($V$). $W$ is a feature extractor applied on the input and returning a $K$-bit index i.e. a function $W : \mathbb{R}^{D_{in}} \to \{0,1\}^K$. The computed code-word is then used to select the output representation form the voting table $\boldsymbol{V} \in \mathbb{R}^{2^K \times D_{out}}$:

**Bit functions** - $K$ simple bit functions are used to compute the code-word $W(x; \Theta)$, each computing a single bit. Each bit function compares a single feature to a threshold:

$$w^k(x, \Theta^k) = Q(u^k x - th^k) \tag{1}$$

Where $\boldsymbol{u}^k \in \mathbb{R}^{D_{in}}$ is a 1-hot vector, addressing a single feature from $\boldsymbol{x}$. The learned parameters in the bit function are $\Theta^k = [\boldsymbol{u}^k, th^k]$. Notice that the output of the above comparison is a scalar, hence an Heaviside function $Q(\cdot)$ is applied to produce the needed output: $w^k(\boldsymbol{x}, \Theta^k) \in \{0,1\}$.

**Ferns** - ferns are similar to standard decision trees, but with one important difference - the queries in each node of the same level are fixed, i.e the choice of $w_i$ does not depend on the results of $w_j$ for $j < i$ . Fern is a common machine learning algorithm, and is mostly known as the base predictors of the CatBoost model (Prokhorenkova et al., 2017). The computed code-word is denoted by

$$W(x; \Theta) = (w^0(x, \Theta^0), ..., w^{K-1}(x, \Theta^{K-1})) \tag{2}$$

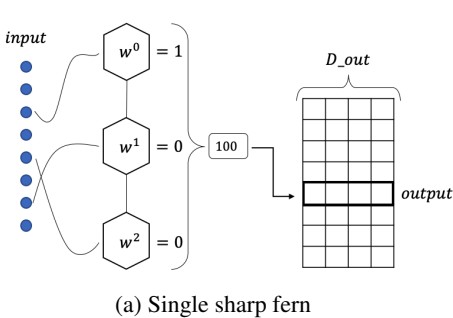

(a) Single sharp fern

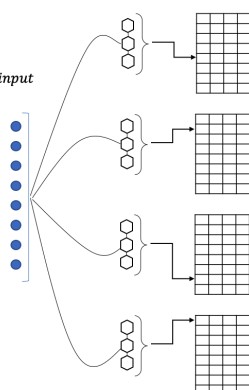

(b) Ensemble of sharp ferns

Figure 1: Illustration of the fern sharp graph including the bit functions and the voting tables. **a** - the bit functions operates on a single input features. Bits are concatenated into single word, pointing to a row in the voting table. The output is a $D_{out}$ dimensional vector. **b** - the voting tables' outputs are summed across ferns.

Since each bit function outputs a single bit, the code-word is a $K$-bit word in a binary expression.

**Voting table** - The computed code-word is then transformed to a numeric value and used as an index to select the $D_{out}$-dimensional output representation vector from a voting table $V$:

$$\hat{h}(x) = V[W(x, \Theta), :] \tag{3}$$

Notice that in the hard version of ferns, $W$ points to a single row in the voting table $V$ and hence the output $\hat{h}(x)$ is a vector. The scheme of a complete fern, including $W$ and $V$ is illustrated in Figure 1a.

**Fern ensemble** - Instead of using a single fern, the S-HTE layer consists an ensemble of $M$ weak learners $\{W_m, V_m\}_{m=1}^{M}$. Ferns outputs are summed to get the layers' final output (see Figure 1b):

$$y = \Sigma_{m=1}^{M} \hat{h}_m(x) \tag{4}$$

## 3.2 DIFFERENTIABLE FERNS

In section 3.1 we describe the basic layer of the deep S-HTE architecture. However, In order to enable end-2-end optimization using gradient based learning, we need to softener the hard components:

**Learnable bit functions** - Instead of forcing bit functions to focus on pre-defined features, we allow them to learn which feature are relevant. In Popov et al. (2019), the entmax (Peters et al., 2019) function was used to choose the features for each bit function. However, as Popov et al. (2019) reported, the entmax function is slow, and though it achieved the best results, it has a small margin from the other feature selection options (e.g softmax, and sparsemax (Martins & Astudillo, 2016)). Hence, we drop the 1-hot vector $u^k \in \mathbb{R}^{D_{in}}$ from Eq. 1, and replace it with a weights vector as follows:

$$u = SoftMax_\beta[B] \quad u_j = \frac{exp(\beta \cdot b_j)}{\Sigma_{i=1}^{D_{in}} exp(\beta \cdot b_i)} \tag{5}$$

Here, $B$ are the learnable parameters that represent the contribution of each input feature. Now, the output is continuous and differentiable. Notice that the output of these bit function are linear combination of the input with respect to the weights in $U$. At the beginning of the training process, the weights will be set randomly, allowing multiple features to contribute, but as the training progresses, we want the bit function to focus on a specific feature for the decision (for each of the bit functions in each fern). Hence we will anneal the temperature parameter $\beta$ from 1 to $\beta >> 1$.

**Soft ferns** - As mentioned, since the output of a bit function (see Eq. 1) is a scalar, the Heaviside function is used. Since $Q(x)$ is not continuous in $x = 0$ and its gradient equals to zero everywhere else, the use of this function is inappropriate for gradient based learning. Hence, a smooth linear sigmoid is used instead:

$$q(x; t) = \begin{cases} 1 & x > t \\ \frac{x+t}{2t} & -t < x < t \\ 0 & x < -t \end{cases} \tag{6}$$

This function behaves like the Heaviside function for values far from 0 but is linear in the section $[-t, t]$ and hence has a gradient in it. When $t \to 0$, $q(x, t)$ approaches the Heaviside function. Thus $t$ can be considered as a hyperparameter controlling the smoothness of the linear sigmoid. In addition, this function has the property $q(x; t) + q(-x; t) = 1$. $q(w^k(\boldsymbol{x}; \Theta^k); t)$ can be interpret as a 'soft' bit, with $q(w^k)$ estimating the probability of the bit to be 1 and $q(-w^k)$ the probability to be 0. As stated above, the word calculator $W$ maps an example $\boldsymbol{x} \in \mathbb{R}^{D_i}$ to a single code-word, or equivalently into a (row) one-hot vector in $\mathbb{R}^{2^K}$. Denote the word calculator viewed as a function into $\mathbb{R}^{2^K}$ by $\vec{W}$, We now extend it to a soft word calculator $\vec{W}^s : \mathbb{R}^{D_i} \to \mathbb{R}^{2^K}$, assigning each possible word a probability-like value. For each code-word $b \in \{0, ..., 2^K - 1\}$, the activity level is defined by

$$\vec{W}^s(\boldsymbol{x}; \Theta)[b] = \prod_{k=1}^{K} q(s(b, k) \cdot w^k(\boldsymbol{x}; \Theta^k); t) \tag{7}$$

Where $s(b, k)(-1)^{(1+a(b,k))}$ is the sign of the $k^{th}$ bit, with $a(b, k)$ denotes the $k^{th}$ bit of the code-word $b$ in the standard binary expansion. $s(b, k)$ is 1 if bit $k = 1$, and $-1$ if bit $k = 0$. The probability-like activity level of a word $b$ is defined as the product of the 'probability' of its single bits. For single hard fern, its output can be written as the product $\vec{W} \cdot V$. The soft fern is the natural extension $\vec{W}^s \cdot V$. Here instead of taking a single output from the table $V$, the Soft-fern outputs a weighted word activation based on the activity levels of the word indices. Furthermore, in our implementation, we may allow only $B$ number of indices to be non-zero for computational complexity considerations.

**Annealing mechanism** - The sparsity of the soft word calculator is controlled by the parameter $t$ of the linear sigmoid in Eq. 6, that acts as a threshold. When $t$ is large, most of the bit functions are 'ambiguous', i.e. not strictly 0 or 1, and the output will be dense. Thus the parameter $t$ controls the output sparsity level. In particular, as $t \to 0$ the bit functions become hard and $\vec{W}^s$ converges toward a hard fern with a single active output word. While we need dense flow of information and gradients in the training phase, fast inference requires a sparse output. Hence, we use an annealing schedule mechanism, such that $t$ is initiated as $t \gg 0$, set to allow a fraction $f$ of the bit functions values to be in the 'soft zone' $[-t; t]$. The value of $t$ is then gradually lowered to achieve a sharp and sparse classifier towards the end of the training phase.

### 3.3 MOVING TO A DEEP LEARNING S-HTE ARCHITECTURE

The above section describes a single layer of the deep S-HTE architecture. Similar to other deep learning schemes, we can stack multiple such layers, where the input of layer $j$ is the output of layer $i$ for $j > i$. The number of output features (i.e $D_{out}$) is arbitrary and should be tuned similarly to tuning the number of neurons in a standard MLP. It is worth mentioning that other similar methods, e.g (Popov et al., 2019; Zhou & Feng, 2017) usually use $D_{out}$ = *number of classes* (for Popov et al. (2019) this is the case due to the computational burden of increasing $D_{out}$). By allowing arbitrary $D_{out}$ size, we enable better representation learning. The final prediction will be the output of the last layer and we train our model end-2-end with backpropagation. Our architecture is best presented in figure 2.

**Adding deep learning machinery** - In the last several years, different regularization methods were reported to achieve significant improvements for deep networks (Zhang et al., 2017; Kadra et al., 2021). In this paper, since we are using a deep model, we exploit these regularization techniques. First, We experimented with different models architectures and found that using a ResNet-like structure (i.e the input to layer $j$ will be the sum of the output of layer $i$ with the input of layer $i$, where

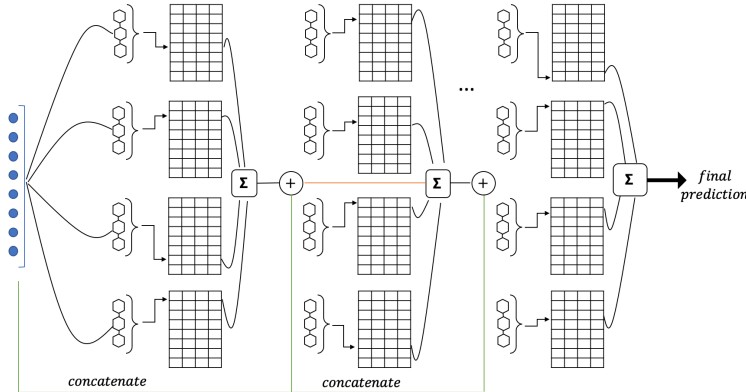

Figure 2: Deep S-HTE architecture - multiple layers scheme. Each voting table output is summed to get the layer's output. This output in then concatenated with the input example (green lines), and the outputs of each layer is summed across layers (red line).

$j > i$) along with the concatenation of the input example, yields the best results. Second, we add a batch normalization layer, a weight decay factor and use different learning rate schedulers.

## 4 COMPUTATIONAL COMPLEXITY COMPARISON

We analyze computational complexity based on the number of expected operations in S-HTE and other relevant models: MLP, FT-Transformer (Gorishniy et al., 2021), NODE (Popov et al., 2019) and decision tree ensembles. For deep methods, we assume for simplicity that all layers have the same internal representation dimension $D$. For operation count we consider multiply-accumulate as a single operation, and denote by $I$ the number of instructions required for a bit function computation. Typically $I$ is $2 - 3$ operations: comparison to a threshold, a *shift left* operation, and an *or* operation to store the computed bit.

**S-HTE complexity.** A standard S-HTE uses a single $B$-bits index for inference in each fern. A single fern inference hence requires the index computation with $BI$ operations and then addition of the relevant table row to the output representation with $d$ operations. The total inference cost for $L$ layers of $M$ ferns each is

$$LM(BI + D) \tag{8}$$

In this analysis we neglect the cost of bypass additions ($LD$ operations) and classification head computation ($DK$ operations, with $C$ the number of classes).

When ferns with $P$ split-able bits are considered, the number of active words used in inference is $2^P$. For each such word index computation required $P$ bit operations, and probability computation requires $P$ multiply operations. The total computation of the $2^P$ words is hence $2P \cdot 2^P$ operations. Adding the relevant $2^P$ table rows with the corresponding weights requires $2^P D$ operations, so the total inference cost is

$$LM(BI + 2^P(2P + D)) \tag{9}$$

In our multi-word predictors, we consider modest numbers of ferns $M \leq 100$ and split-able bits $P = 2, 3$.

**NODE.** The NODE architecture is similar to our multi-word predictors, but with the crucial difference of using all bits as split-able bits, i.e. $P = B$. Since the dependence on $P$ is exponential, this lead to a dramatic complexity difference.

**Standard MLP.** The first layer in an MLP with $D_{in}$ input includes $D_{in}D$ operations and the rest of the layers are with $D^2$ operation count. the total complexity (ignoring bypass and the classification head) hence has a quadratic dependency on $D$

$$(L - 1)D^2 + D_{in}D \tag{10}$$

**FT-Transformer.** The architecture suggested in Gorishniy et al. (2021) includes a feature tokenizer, folllowed by $L = 3$ attention layers and a prediction head. Encoding by the tokenizers requires $D_{in}D$ operations. The attention model in each layer requires $3D_{in}D^2$ operations for $Q, V, K$ matrices creation, $D_{in}^2D$ for computing the $D_{in}$ key-query matrices, and $D_{in}^2D$ for computing the output messages. The MLP following each attention module also requires $D_{in}D^2$ operations, so the total per $L$ layers (ignoring the negligible $O(D_{in}D)$ operations) is

$$L(3D_{in}D^2 + 2D_{in}^2D) \tag{11}$$

**Shallow ensemble trees.** A single tree in such ensembles is extremely fast, requiring $BI$ operations for bit function compute followed by $C$ additions to gather the class votes. However, the total inference cost is $M(BI + C)$ and the number of trees required by such methods is very large due to their shallow architecture and step-wise optimization. Specifically, Thousands of trees are typically required to reach high accuracy, while S-HTE typically uses several hundreds.

In section 5 the speed and accuracy of S-HTE is compared to the above-mentioned methods with their real implementation parameters (see table 2).

## 5 EXPERIMENTS

**Experimental settings.** The S-HTE is fully implemented in PyTorch and is publically available[2]. The training code was developed to utilized the GPU, and models were trained on a 1080TI Nvidia-GPU. We used ADAM (Kingma & Ba, 2014) as optimizer, and enabled different learning rates for the ferns' and the voting tables' parameters - voting table parameters may have a high learning rate as opposed to word calculators, where each gradient-step can dramatically effect the function's output. We also report how we set hyper-parameters' values in our experiments, as well as the model's architecture and weights initialization.

**Training procedure.** Similar to DNNs, the S-HTE architecture is trained end-to-end with mini-batch SGD. We used batch sizes related to the dataset size (following (Gorishniy et al., 2021)) and trained the network for 80 epochs. For classification tasks, we minimize the cross-entropy loss function, and for regression tasks we minimize the mean-square-error. We save the final trained model (in which all the ferns are sharp and only a single word is used for the voting table) and the best model on a held-out validation set. The latter is usually not completely sparse and hence may achieve better results.

**weights initialization.** Similar to NODE (Popov et al., 2019), we initialize the ferns feature selection matrix uniformly $\boldsymbol{u}_{i,j} \sim U(0, 1)$ for $i \in 1, .., K$ and $j \in 1, ..., M$. The thresholds values are initialized with zeros. The weights in the voting table $\boldsymbol{V}$ are initialized with a standard normal distribution $\boldsymbol{V}^m \sim \mathcal{N}(0; 1)$. The softmax temperature parameter $\beta$ is initialized as 1 and annealed towards 30 at the end of training. The smooth step function threshold ($t$ in Eq. 6) is controlled by an additional parameter termed $\rho$ - this parameter controls the percentage of examples with non-zero gradient (see section 3.2 for more details). We set $\rho = 0.99$ at the beginning of training, and slowly decay it towards 0.

### 5.1 COMPARISON TO THE STATE-OF-THE-ART

**Dataset.** We've experimented with six tabular datasets: Adults Income (AD), ALOI (AL), Jannis (JA), Higgs-small (HI), California housing (CA), and Year (YE). These datasets were all taken from Gorishniy et al. (2021) open-source library and were already pre-processed. For multi-class datasets (AL, JA) we report the accuracy on the test set, and for binary classification problems (AD ,HI), we report the Area Under the Curve (AUC) of the ROC curve. For regression problems (CA, YE) we report the Root-Mean-Square-Error. A detailed description of the datasets can be found in the appendices.

**Methods.** We compare our model to the following baselines:

- **CatBoost.** a GBDT implementation which uses ferns as the computational component.
- **XGBoost.** Similar to CatBoost, with RDTs as the computational component.

---

[2]https://anonymous.4open.science/r/HTE_CTE-60EB/

Table 1: Comparison to the state-of-the-art in terms of accuracy. All comparative results are taken from the recent research on tabular data (Gorishniy et al., 2021). Dataset details can be found at the appendix A. Results are reported for S-HTE which has one active word per fern, and 'best S-HTE', which has several active words (1-10, stated in the parenthesis).

|  | AD $\uparrow$ | AL $\uparrow$ | JA $\uparrow$ | HI $\uparrow$ | CA $\downarrow$ | YE $\downarrow$ |
|---|---|---|---|---|---|---|
| CatBoost$_d$ | 0.797 | 0.946 | 0.721 | **0.724** | **0.430** | 8.913 |
| XGBoost$_d$ | 0.775 | 0.925 | 0.721 | 0.705 | 0.463 | 9.446 |
| MLP | 0.796 | **0.954** | 0.719 | 0.721 | 0.494 | 8.861 |
| TabNet | 0.796 | **0.954** | 0.724 | 0.717 | 0.513 | 9.032 |
| FT-Transformer$_d$ | 0.799 | **0.953** | **0.725** | 0.723 | 0.464 | 8.820 |
| NODE (flat) | N/A | 0.918 | N/A | N/A | 0.464 | N/A |
| NODE (optimized) | 0.791 | N/A | **0.726** | **0.724** | N/A | **8.774** |
| S-HTE | 0.801 | 0.948 | 0.672 | 0.695 | 0.518 | 9.253 |
| best S-HTE | **0.817** (4) | **0.953** (4) | 0.707 (10) | 0.708 (6) | 0.474 (1) | 9.210 (2) |

- **MLP.** Simple feed forward network, with several standard Linear-ReLu-Dropout layers, with dropout regularization.

- **TabNet.** Sequential attention based model, enabling feature selection for each decision step.

- **FT-Transformer.** Transformer based model designed to handle tabular datasets.

- **NODE.** Dense differeniable ensemble of ferns (this method is the closest to the ours). NODE has two variants - a flat model (i.e. single layer of ferns) and a optimized model in which several layers are used. Both variants were used as described in Popov et al. (2019).

We used the same parameters for each of the datasets ($L = 3$, $M = 100$, $K = 7$, $D = 40$), with the exception of AL dataset which has 1000 output classes (we used $L = 4$, $K = 8$, and $M$, $D$ remained the same).

The results are summarized in Table 1. The comparative methods' results were taken from Gorishniy et al. (2021), which published a comprehensive analysis over several contemporary deep learning models for tabular data. These results advocate the S-HTE model as a competitive method in terms of accuracy, specifically for the AD, AL, and CA datasets. We report the results of the fully-sparse model, which contains completely sharp ferns, and the best trained model on a held-out validation set, which is not necessarily fully sparse. For the latter, the average number of active words in a fern is reported next to the results on each dataset in parenthesis. In comparison to NODE, NODE works better for JA, HI, CA, and YE datasets, with a small margin compared to the best S-HTE, while S-HTE is superior on AD and AL.

Table 2: Comparison to the state-of-the-art in terms of computational complexity. The equations from section 4 are used to compute the algorithms operation count. The numbers represents thousands of operations.

|  | AD | AL | JA | HI | CA | YE |
|---|---|---|---|---|---|---|
| CatBoost$_d$ | 40 | 2036 | 44 | 40 | 38 | 38 |
| XGBoost$_d$ | 40 | 2036 | 44 | 40 | 38 | 38 |
| MLP | 602.38 | 26.21 | 31.21 | 34.42 | 233.68 | 1155.43 |
| FT-Transformer$_d$ | 16025.86 | 82575.36 | 60466.18 | 12536.83 | 5431.3 | 46033.92 |
| NODE | 24147.97 | 66340.86 | 2134.017 | 2202.94 | 935.94 | 2002.94 |
| S-HTE | 18.3 | 17.9 | 18.3 | 18.3 | 18.3 | 18.3 |
| best S-HTE | 124.93 | 4163.58 | 561.15 | 159.74 | 75.77 | 159.74 |

As discussed in section 4, we compared the S-HTE to the state-of-the-art methods for tabular data by the number of expected operations. Table 2 shows the number of operations (in thousands) used by the analyzed methods on the examined datasets (Based on hyper parameters taken from the official repository of (Gorishniy et al., 2021)[3]). As can be seen, S-HTE has the lowest number of

---

[3] https://github.com/yandex-research/rtdl/tree/main/output

Table 3: Accuracy as a function of S-HTE model depth on the Adult income dataset. As in the other experiments, a 3-layers model works best on this dataset. As can be seen, deep models provide accuracy gains over shallow ones (for example, the 4 layers model is better than the 2 layers one).

| Model Variant | Final Model Results | $\Delta$ |
|---|---|---|
| 1-Layer | 0.754 | - |
| 2-Layers | 0.757 | +0.4% |
| 3-Layers | 0.772 | +2.4% |
| 4-Layers | 0.769 | +1.9% |

Table 4: Ablation study over the annealing components. We test the effect of feature sparsity in a bit function (using the $\beta$ parameter) and of word sparsity in a fern (using the $\rho$ parameter). Results are stated on the Jannis dataset, for which the gap between dense and sparse results are significant.

| Model Variant | Final Model Results | $\Delta$ |
|---|---|---|
| Without annealing | 0.667 | - |
| Annealing of $\beta$ only | 0.693 | +2.6% |
| Annealing of $\rho$ only | 0.648 | -1.9% |
| Annealing both $\rho$ and $\beta$ | 0.672 | +0.5% |

operations, often by orders of magnitude. Note that these results are obtained without an extensive hyper-parameter tuning study, from which S-HTE can probably gain more.

## 5.2 Ablations Study

**From Flat to Deep Models.** To examine the benefits of deep representation learning, we compare the results of flat model against $L = \{2, 3, 4\}$. In this experiment we fixed the rest of the model's parameters - $M = 50$, $K = 7$, $D = 40$. The accuracy of the final trained models (i.e. containing only sharp ferns) are presented in Table 3. As the results indicate, using deeper models improved the results by up to $\sim 2.4\%$. Similar experiments with the other datasets show that for all datasets except AL $L = 3$ yielded the best results.

**Annealing Mechanism Ablation Study.** As discussed in section 3.2, there are two annealing mechanisms in the S-HTE architecture - annealing the $\rho$ parameter for sparse usage of active words, and annealing the $\beta$ parameter for obtaining feature-specific bit functions. We examine the effect of these mechanisms by disabling them. In these experiments we used the Jannis dataset, on which a large accuracy gap between a complete sparse model (our S-HTE), and a complete dense model (the optimized version of NODE) was observed. We use a 3-layered model, with 100 ferns in each layer, and 7 bit functions in each fern. Surprisingly, the best result was obtained using a partially annealed version of the S-HTE model - annealing the $\beta$ parameter only, keeping $\rho = 0.99$. This means using multiple active words is beneficial, but each bit is these words should depend on a single feature. Upon reflection, this result has a natural explanation - tabular data features are typically heterogeneous and different in nature (i.e. they describe different physical quantities, not similar entities as pixels in different locations). Hence considering each feature in isolation is a reasonable strategy, and enforcing such feature isolation is providing a useful prior.

## 6 Conclusion

In this work, we introduced the S-HTE architecture - a sparse deep learning model for tabular data trained end-to-end with back-propagation. S-HTE uses ferns as the basic computing component, with a single active word for each fern, making is extremely efficient and particularly applicable for low-end CPU-based devices. Using a significantly lower operation count, accuracy results obtained by S-HTE are often comparable to state of the art architectures, as measured on a recent benchmark. Ablative studies show S-HTE networks earn from depth, and that computing sparse bit functions relying on single features is beneficial to both accuracy and efficiency. In the future we plan to experiment with data-specific hyper-parameter tuning, and incorporation of recent advances in regularization techniques, which were found to have a significant effect on MLP performance (Kadra et al., 2021).

## 7 ETHICS STATEMENT

As far as we are aware, there are no ethic concerns in this work.

## 8 REPRODUCIBILITY

Our code is publicly available at `https://anonymous.4open.science/r/HTE_CTE-60EB/`. All of the datasets used in this paper were taken from a previous published work (https://github.com/yandex-research/rtdl/tree/main/output). The datasets were preprocessed in advanced (by the above repository owners), and we kept the same train-validation-test splits.

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

# A  DATA

**Datasets**

Table 5: Datasets description

| Name | #Train | # Val. | #Test | #Features | Task | #Classes | Batch size |
|------|--------|--------|-------|-----------|------|----------|------------|
| Adults Income | 26048 | 6513 | 16281 | 14 | Classification | 2 | 256 |
| Jannis | 53588 | 13398 | 16747 | 54 | Classification | 4 | 512 |
| Higgs-small | 62752 | 15688 | 19610 | 28 | Classification | 2 | 512 |
| ALOI | 69120 | 17280 | 21600 | 128 | Classification | 1000 | 512 |
| California Housing | 13209 | 3303 | 4128 | 8 | Regression | - | 256 |
| Year | 370972 | 92743 | 51630 | 90 | Regression | - | 1024 |

