# OpenReview forum: "Sparse Hierarchical Table Ensemble"
_ICLR.cc/2022/Conference — ICLR 2022 Submitted_

### Official Review · Reviewer_dw2A · 2021-11-03

**Correctness:** 3
**Technical Novelty And Significance:** 2
**Empirical Novelty And Significance:** 2
**Recommendation:** 5
**Confidence:** 4

**Main Review:**

Novelty: the proposed approach can be considered as an extension of the NODE (Popov et al. 2019) with annealing mechanism. From my point of view, this requires a minor modification to the original algorithm: softening of Heaviside function was done differently with annealing hyperparameter taken into account. Furthermore, the annealing mechanism is relatively old trick that is commonly used in literature to get sparsity. Other technical details such as using deep architectures, concatenating inputs/outputs, etc. were already done in the mentioned paper. Therefore, I think that the methodological contribution is not significant.

Experiments: evaluations suggest that the proposed method perform slightly worse than NODE but more efficient and I agree with that. However, by looking at Table 1, I can see that the performance of CatBoost/XGBoost are quite similar. In some cases they even notably outperform S-HTE (e.g. Higgs-small, California Housing) while still being very efficient (asymptotically). Moreover, they are highly optimized for different hardware settings (CPU/GPU) and can benefit from parallel processing. Given all of these, I would suggest to simply use CatBoost or XGBoost rather than the proposed approach. Probably, the proposed method can improve significantly for non-tabular data (e.g. images and texts) where correlation of features matter where representation learning matters (and traditional boosting machines do not perform well).

Other comments/questions:
- annealing is applied to the Heaviside function but not for the bit functions. Does it mean that each bit function uses a linear combination of all features or they are also sparsified in some way? In any case, the resulting oblivious trees are effectively oblique ferns. They are more powerful class of models rather than traditional axis-aligned trees but they are typically less interpretable.
- "the computational component in XGBoost are Random Decision Trees (RDTs)". What do you mean by random trees? As far as I know, XGBoost use traditional CART style decision trees which are not random.

**Summary Of The Paper:**

Authors propose a variant of differentiable multilayer tree ensembles with annealing mechanism. They use a specific types of trees, called ferns, which are constrained to have the same boolean function for all nodes at the same level. Such a tree representation can be useful to effectively reduce a decision tree into a table lookup. The proposed model can be considered as multilayer ferns where the output of the given fern is used as input to the next one and the entire model is trained end-to-end. Additionally, the input from the previous layer is stacked with output as in ResNets. Training is done by softening non-differentiable parts of ferns to allow gradient based optimization. Finally, annealing mechanism is used to obtain sparse connections.

**Summary Of The Review:**

Based on major concerns regarding technical novelty and experiments, I am leaning towards rejection of this paper.

---

### Official Review · Reviewer_vA5d · 2021-11-04

**Correctness:** 3
**Technical Novelty And Significance:** 1
**Empirical Novelty And Significance:** 1
**Recommendation:** 1
**Confidence:** 5

**Main Review:**

Overall the paper writing is clear. The related work is also good and comprehensive although there is one similar work they miss (see below).

I have several major concerns. Notably:
1. The proposed idea of making soft to hard operations has been proposed in [1], although [1] focuses on making NODE as a Generalized Additive Models (GAMs) and not specifically about the lower computational cost.
2. Although this paper focuses on lowering the computational cost and is suitable for the lower-end CPU, there is no empirical measurement and computing time to support it. Also, in nowadays CPU and GPU with strong parallel processing architecture, the proposed sparsity might not enjoy any speed up in the real world since either dense or sparse vector will require a dot product in PyTorch.
3. The claimed contribution is also unclear when no experiment of speed up is compared to traditional tree-based packages. They claim that S-HTE enjoys the deep-learning representation and thus only require hundreds of trees while traditional trees require thousands of trees, but this claim needs to be backed up in the experiment. In fact, sklearn GBT default tree number is 100, which is contradictory to what authors claim.
4. The datasets selected are from Gorishniy et al. 2021. Why don't you experiment on NODE's datasets since NODE is the closest one to your method? This makes readers suspicious of the accuracy of the method.
5. The ablation study is only conducted on 1 single dataset which is very unconvincing.
6. Authors should also compare with other sparsification technique for neural network like [2].

[1] NODE-GAM: Neural Generalized Additive Model for Interpretable Deep Learning, https://arxiv.org/abs/2106.01613
[2] Variational Dropout Sparsifies Deep Neural Networks: https://arxiv.org/abs/1701.05369

**Summary Of The Paper:**

This paper modifies from a differential-tree architecture (NODE) for tabular data that has lower theoretical computational complexity to be suited for lower-end CPU devices. Specifically, they gradually anneal the soft operations in NODE to hard operations that sparsify the operations and thus has lower theoretical computation. They find this sparsification, with other modifications like summing across the outputs for each layer before passing to the next layer, can enjoy much smaller operation counts. They show that their performance is somewhat comparable to other commonly used methods like CatBoost, TabNet and NODE. And they do an ablation study to confirm sometimes deeper layers improve the performance and sparsification also leads to higher accuracy.

**Summary Of The Review:**

Overall this paper proposes a similar idea to previous work [1] that also sparsifies the NODE without mentioning it. There is no empirical measurement of the claimed speed up, and the selected datasets are not compared to the datasets used in NODE, and no baseline including sparsified neural network like [2]. The accuracy results are somewhat lower than CatBoost in 4 out of 6 datasets.
To summarize, there is very little novelty in the architecture, no empirical time measurement to back up its claim, and the datasets selected are not comprehensive, and some missing important baselines. So I recommend rejection for this paper.

---

### Official Review · Reviewer_M9FK · 2021-11-05

**Correctness:** 3
**Technical Novelty And Significance:** 2
**Empirical Novelty And Significance:** 2
**Recommendation:** 5
**Confidence:** 3

**Main Review:**

STRENGTHS

- The paper is clear about the relation between the proposed method and the previous works. This is very helpful for understanding the paper.
- The complexity analysis is helpful for understanding the computation efficiency of the proposed method.


WEAKNESSES

- The computation cost comparison is in numbers of operations, instead of the wall time on fixed hardware. This may be unfavorable for the dense methods, since the dense operations are usually easier to be accelerated on modern hardware, especially GPU.
- The comparison on the datasets does not show clearly the proposed method is better. If efficiency is the main advocacy, it may be better to put the accuracies in the context of the computation wall time.
- The usage of a validation set for parameter tuning may not be unfair for the other methods. Essentially, the validation set is used as a training set for the proposed method. From the paper, it is not clear whether the other methods use the validation set for training as well.
- The ablation on layer number of the proposed model in Table 3 shows that more layers only add the performance marginally. This is different from the observation in deep convolutional networks that deeper models usually perform much better. The present ablation study is undermining the motivation that we want to go deeper with the proposed method.

**Summary Of The Paper:**

The paper constructs Sparse Hierarchical Table Ensemble (S-HTE) based on oblivious decision trees for tabular data processing. The motivation is to benefit from the expressiveness of multi-level processing based on a differentiable framework. The paper dives into how ferns work and the complexity of the new design. The experiments are conducted on the commonly used tabular datasets.

**Summary Of The Review:**

Despite the theoretical efficiency advantage of the proposed method, there are still remaining questions as mentioned above. I am leaning towards rejecting given those questions.

---

### Official Review · Reviewer_2rMh · 2021-11-07

**Correctness:** 3
**Technical Novelty And Significance:** 3
**Empirical Novelty And Significance:** 2
**Recommendation:** 5
**Confidence:** 2

**Main Review:**

First of all, I am not an expert of this specific problem (tabular data prediction and differentiable decision tree learning), although I published papers on decision trees before.

Strength:
- The paper is clearly written. Although I am not working in this field, I am able to follow the paper easily.
- From the introduction and related work of the paper, technically, this paper seems to have addressed some challenges that previous works have not addressed before (for detailed explanations, see "Differentiable decision trees based architectures." in the related work section).

Weakness:
- The performance is not so impressive (esp. in comparison to CatBoost which is also quite fast).
- The results are not reported with an error bar. Without the error bar, some of the designs, such as the ablation of having deep architecture, are not quite convincing given the limited improvement by increasing L from flat to 4. Especially, the performance drops a bit when increasing L from 3 to 4. I wonder whether deep is really necessary for such methods.

**Summary Of The Paper:**

This paper studies the prediction problem on tabular datasets and proposed a differentiable multi-layer fern-based architecture. The training and inference algorithms are provided, and the computational complexity has been analyzed. The selling point is that the approach is computationally efficient on CPUs and can learn sparse representation. The prediction performance is on-par with state-of-the-art methods.

**Summary Of The Review:**

In short, the paper has some merits in comparison to the literature of differentiable decision trees. However, its performance is not so strong and some experiment results are not solidly justified. Nonetheless, I feel the cited papers by this submission are also somewhat incremental to their literature.

---

### Decision · Program_Chairs · 2022-01-20

**Decision:**

Reject

**Comment:**

The submission introduces the sparse hierarchical table ensemble (S-HTE), based on oblivious decision trees for tabular data. The reviewers acknowledged the clarity of the presentation and the importance of the computational complexity analysis. However, they also raised concerns regarding the novelty of the proposed method and the significance of the results compared to competing methods (e.g., CatBoost). Given the consensus that the submission is not ready for publication at ICLR, I recommend rejection at this point.